# Pan for Gold: Generalization Emerges Naturally through Stochasticity

## Abstract

Training a deep model is fundamentally about reducing loss, and we often believe that a "good model" is one that trained with a "good loss." This paper investigates that belief. We show that even when learning with unstructured, randomized labels, models can still discover generalized features. We propose that generalization in deep learning is not about learning the structure of data through a well-structured loss, but rather a process akin to "pan for gold," where gradient descent shakes through the function space, naturally stabilizing useful features. To support this, we present quantitative and qualitative experimental evidence, and introduce the Panning through Unstructured Label (PUL) algorithm. We demonstrate its effectiveness across various fields, showing improvements in unsupervised domain adaptation, state-of-the-art performance in object discovery, and its ability to mitigate massive attention issues. Finally, we offer a new interpretation of existing deep learning assumptions, challenging the conventional beliefs in the field.

## 1 Introduction

Most modern deep learning involves reducing a certain loss in an overparameterized model using SGD. Since SGD tries to minimize a given loss, this inherently reflects the assumption that *the lower the loss, the better the model*. The loss serves as a metric for how well the model captures the structure in the data during training. A lower loss implies that the model has learned more about the data structure from the training data. For instance, in image classification problems, we define a classification loss that reflects the structure of image-category pairs (Deng et al., 2009) or use a consistency loss (Chen et al., 2020; He et al., 2020; Grill et al., 2020; Caron et al., 2021b) that encodes our belief that two transformed versions (e.g., through cropping or rotation) of the same image should represent the same object. In any case, this process rests on two key ideas: 1) SGD reduces the given loss, and 2) meaningful reduction requires a well-defined structure.

This paper deals with the following question: Is structure itself the essence of a 'good' result? We approached this question with a proof by contradiction: if structure were essential for producing good results, removing the structure should lead to poor results. We completely removed the structure from the learning process by randomizing the class labels, and found that the model actually was able to learn from data despite the complete randomization and even performed better from a generalization perspective. Based on these observations and mathematical analysis, we propose a hypothesis about the driving force of generalization in deep learning. We argue that the success of deep learning is not due to the encoding of human beliefs, but to the stochasticity of SGD and diverse exploration of the loss surface of overparameterized models throughout the learning process. The process can be compared to panning for gold: just as randomly shaking a pan sifts out all the impurities except for the gold, there is a very strong shaking in the functional space throughout the learning process which automatically shakes out the noisy features and leaves the 'good' features. In doing so, we show that the unstructured random labels leaves more 'gold'. In this paper, we present strong experimental evidence supporting our bold hypothesis. Furthermore, we present an algorithm that exploits our hypothesis to dramatically improve performances on existing deep learning problems reaching state-of-the-art performance in some areas in a very simple way.

To analyze this observation, we view the development of deep learning in the functional space. First, we investigate the condition of functional descent necessary for loss reduction. Alongside this analysis, we observed a swing phenomenon, where the output of the neural network fluctuates dra-

matically during the early stages of training. We explain this phenomenon using the Neural Tangent Kernel (NTK) (Jacot et al., 2018). Based on our observations and analysis, we propose the "pan for gold" hypothesis, which suggests that as learning progresses, the swings of the function diminish, highlighting only the important features of the given dataset. To verify our hypothesis of *Pan for Gold*, we conducted several experiments. First, we confirmed the presence of effective functional descent by demonstrating the positive definiteness of the NTK and observing a flattened loss surface along the gradient direction. Next, using Grad-CAM visualizations (Selvaraju et al., 2017), we illustrated that even models trained with unstructured labels naturally focus on meaningful features.

Based on our hypothesis, we proposed an algorithm, Panning through Unstructured Label (PUL) that improves the generalization ability of existing deep learning models by using unstructured labels. First, we demonstrated that PUL enhances performance in unsupervised domain adaptation, where accessing target labels is restricted. By simply assigning random labels to the unlabeled target data, we increased performance in classification and object detection. Moreover, we achieved state-of-the-art (SOTA) performance in the object discovery task within just a few epochs using ResNet50, despite not using any source labels, which made the problem more challenging. These improvements demonstrated that unstructured labels can enhance attention maps. We also observed a significant improvement in the activation distribution and successfully alleviated the widespread attention issues (Xiao et al., 2024), which are a major challenge in the quantization field. The provided results show that our 'gold-preserving' algorithm significantly enhances performance across various domains that require generalization. Lastly, we argued that our methodology not only improves existing algorithms quantitatively but also offers a new perspective on certain phenomena. For example, while the edge-biasing issue observed in XAI methods like Grad-CAM (Selvaraju et al., 2017) is often regarded as a flaw of these methods, we confirmed that it is not a defect, but rather a natural outcome of the model's learning process.

## 2 LEARNING FROM UNSTRUCTURED DATA

### 2.1 CONVENTIONAL BELIEF ON DATA

The goal of deep learning is to learn from data according to structures defined by humans. These structures are shaped by human prior knowledge or established beliefs. They form the essence of the model's learning process. People mathematically conceptualized this belief as 'Energy', and we use the Energy-Based-Models (EDM) (Xie et al., 2022) for developing our idea.

Humans have a belief that data can be categorized based on certain criteria that are used to define input-query pairs. For a given data $x \in \mathcal{X}$, they classify it and assign a structured label $y \in \mathcal{Y}$ according to these assumptions. The supervision energy function $E_{sup}$, which represents the state of the model $f_\theta$, and hence the data, can be defined as follows for an arbitrary distance metric $d$:

$$E_{\text{sup}}(f) \triangleq \mathbb{E}_{(x,y) \sim p(\mathcal{X}, \mathcal{Y})} \left[ d(f_\theta(x), y) \right], \tag{1}$$

where a pseudo-metric in the probability space, such as cross-entropy (CE) is typically used for $d$.

Another key human belief about data is *augmentation-invariance*: the idea that the essence of an image remains unchanged for a slight deformation. This belief is applied in Self-Supervised Learning (SSL) where a set of data augmentations $\mathcal{A}$ is defined, and the model is trained to minimize the energy defined as the output distance between different versions of the same image. The SSL energy $E_{ssl}$ is defined as follows:

$$E_{\text{ssl}}(f) \triangleq \mathbb{E}_{x \sim p(\mathcal{X})} \mathbb{E}_{a_1, a_2 \sim p(\mathcal{A})} \left[ d(f_\theta(a_1(x)), f_\theta(a_2(x))) \right]. \tag{2}$$

In the end, humans have conventionally believed that what matters is how humans structure the data, whose structure is shaped by the underlying human beliefs about the data itself.

### 2.2 CONTRADICTION: LEARNING OCCURS EVEN WITHOUT STRUCTURE

We can verify the validity of this belief through a simple experiment. By assigning random structures to the dataset (essentially applying unstructured labels), we can determine whether the model is able to learn. If the belief is correct, the model should not learn effectively under these conditions.

We trained a model on an unstructured dataset $(\mathcal{X}, \mathcal{Y}_u)$, where the labels lacked any inherent structure. The model was trained until it appeared to have learned sufficiently. We then measured the

Table 1: Classification accuracy of linear probing on CIFAR-10 and SVHN. 'Random Init' denotes initialization with random weights, while '$\mathcal{Y}_u$ trained' refers to initialization using the weights of a model trained on the CIFAR-10 dataset with unstructured labels.

| Dataset | Random Init (%) | $\mathcal{Y}_u$ trained (%) |
|---------|-----------------|------------------------------|
| CIFAR   | 19.60           | 32.51                        |
| SVHN    | 19.31           | 34.19                        |

performance of its encoder. To assess whether the model had truly learned anything meaningful, we applied transfer learning to the frozen encoder.

Table. 1 demonstrates the result that the model learns meaningful knowledge even from unstructured data. Thus, we can infer that the energy-based hypothesis of deep learning fails to explain the secrets of generalization. Now, we need a new hypothesis to understand how learning occurs without inherent structure of data. Where does generalization come from?

## 3 PAN FOR GOLD

We propose a new hypothesis regarding the learning process of deep models, which we call the "pan for gold." We suggest that, akin to sifting for gold in a river, stochastic gradient descent (SGD) inherently filters out unnecessary features as it navigates through the function space of an overparameterized model. Just as water and sand (irrelevant features) pass through a sieve, leaving behind valuable gold (essential features), this filtering occurs naturally, without any active search for useful elements. We argue that this process takes place independently of human-imposed structures or belief systems.

**Overview** Our claim is that model generalization naturally arises when certain conditions (Eq. (3)) are satisfied. In Sec. 3.2, we analyze what these conditions mean in terms of functions, under the idea that because overparameterized models memorize any training dataset, the corresponding loss should decrease in an overparameterized setting. We explore experimentally in Sec. 3.3 that this process can be highly unstable, and analyze it through NTK. From these observations, we deduce the "pan for gold" hypothesis in Sec. 3.4. Finally, in Sec. 3.5, we qualitatively assess whether our hypothesis holds true and if the learning process follows the pattern we propose.

### 3.1 PRELIMINARIES

A deep learning model can be viewed as a function conditioned by parameters, which generates outputs for various inputs. From this perspective, the change in weights during gradient descent can be interpreted as the development of the function over time. By observing how the function evolves over time, we can gain a deeper understanding of the model's learning process.

In this context, the **Neural Tangent Kernel (NTK)** (Jacot et al., 2018) serves as a powerful tool for such functional analysis. NTK models the deep learning model $f(\cdot, \theta)$ as a kernel $k(\cdot, \cdot)$, and by analyzing this kernel, we can perform functional analysis of the learning dynamics. Through this approach, we can mathematically interpret the complex learning process of deep learning models. By analyzing the eigenvalues of the NTK, for instance, we can determine how rapidly the model is changing during training and how efficiently it is minimizing the loss.

### 3.2 FUNCTIONAL GRADIENT DESCENT

In the end, what we know is that an overparameterized deep model can fit to any dataset $(\mathcal{X}, \mathcal{Y})$. But what does it actually mean to fit? In a simple term, it means that the **loss decreases** as the number of epochs increases, *i.e.* $\mathcal{L}(\mathcal{X}, \mathcal{Y}; \theta_{t+1}) < \mathcal{L}(\mathcal{X}, \mathcal{Y}; \theta_t)$. To meet this condition, the following two

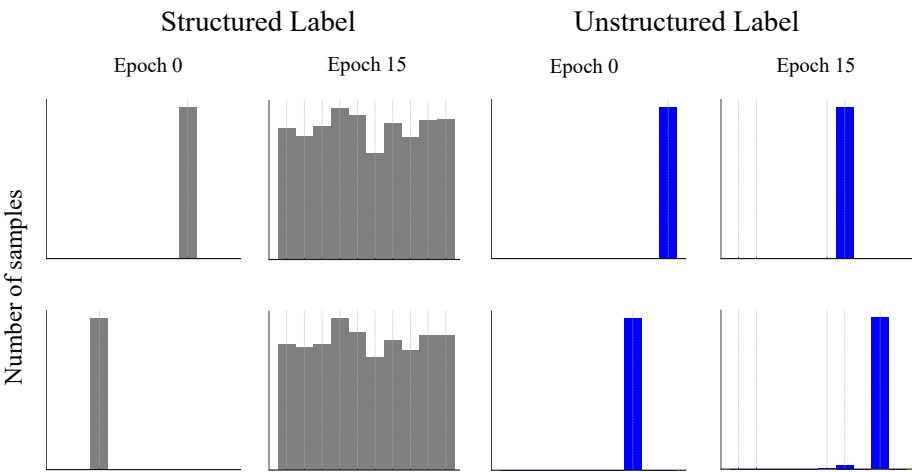

Figure 1: **The swing phenomenon.** For both structured (CIFAR10) and unstructured (CIFAR10 with random label) data, the network outputs concentrated on a single class at an early stage of learning (Epoch 0). We call this 'swing phenomenon' and it lasted longer for the unstructured data (Epoch 15). $x$-axis is the predicted class index (among 10 classes) and $y$-axis represents the number of samples. The first row presents the results after the first iteration, while the second row corresponds to the results after the second iteration.

assumptions are highly likely to be satisfied:

$$\text{A1.} \quad \underbrace{\nabla_\theta^2 \mathcal{L}(\mathcal{X}, \mathcal{Y}; \theta)}_{\text{Hessian } \mathcal{H}} \underbrace{\nabla_\theta \mathcal{L}(\mathcal{X}, \mathcal{Y}; \theta)}_{\text{Gradient } g} \simeq \mathbf{0},$$

$$\text{A2.} \quad f(\mathcal{X}, \theta_t - \alpha \nabla_\theta \mathcal{L}(\mathcal{X}, \mathcal{Y}; \theta_t) + \varepsilon) \simeq f(\mathcal{X}, \theta_t - \alpha \nabla_\theta \mathcal{L}(\mathcal{X}, \mathcal{Y}; \theta_t)). \tag{3}$$

The first condition relates to the complexity of the model's loss landscape, especially in an overparameterized model. Since Gradient Descent (GD) is an algorithm that approximates the model in a locally linear fashion, it simplifies the problem by ignoring the high-order terms that describe the nonlinearity of the loss surface. In reality, however, due to overparameterization, the loss landscape is highly complex and can be highly nonlinear. To ensure the loss to decrease in this setting, it is crucial to account for these higher-order terms, particularly the second-order effects represented by the Hessian $\mathcal{H}$.

To ensure that the GD algorithm drives the model towards a lower loss despite this complex loss surface, the following inequality must be satisfied (Lee et al., 2023):

$$\int_0^1 \nabla_\theta \mathcal{L}(\theta(\tau)) \cdot \nabla_\theta \mathcal{L}(\theta_t) d\tau \approx \int_0^1 \|\nabla_\theta \mathcal{L}(\theta_t)\|_2^2 - \alpha\tau \nabla_\theta \mathcal{L}(\theta_t)^T \nabla_\theta^2 \mathcal{L}(\theta_t) \nabla_\theta \mathcal{L}(\theta_t) d\tau > 0, \tag{4}$$

where we linearly interpolate $\theta_t$ and $\theta_{t+1}$ updated by GD with learning rate $\alpha$ for a continuous time, $\tau \in [0, 1]$, as $\theta(\tau) \approx \theta_t - \alpha\tau \nabla_\theta \mathcal{L}(\theta_t)$. Note that $\theta(0) = \theta_t$ and $\theta(1) = \theta_{t+1}$. To meet Eq. (4), $\mathcal{H} \cdot g = \nabla_\theta^2 \mathcal{L}(\theta_t) \nabla_\theta \mathcal{L}(\theta_t)$ should be small which corresponds to Eq. (3)-A1 (Lee et al., 2023).

Additionally, the second assumption arises due to the stochasticity of SGD. Since SGD is essentially an algorithm that estimates the true gradient $g$, the gradient estimate $\tilde{g}$ obtained through SGD on a minibatch can be viewed in the form of $\tilde{g} = g + \epsilon$ for some noise $\epsilon$. At this point, the goal of SGD, like gradient descent, is to move in the direction that reduces the loss surface. Therefore, even when using $\tilde{g}$, it must function similarly to the actual gradient $g$, as described in Eq. (3)-A2.

**Interpretation of Eq. (3)** If the two assumptions hold, it implies that, independent of any label $\mathcal{Y}$, SGD itself causes the model to learn certain characteristics. As gradient descent progresses, the model $f$ naturally distinguishes the direction of gradient descent in a unique way. Through this process, the model develops robustly against the noise that arises during training. The most notable

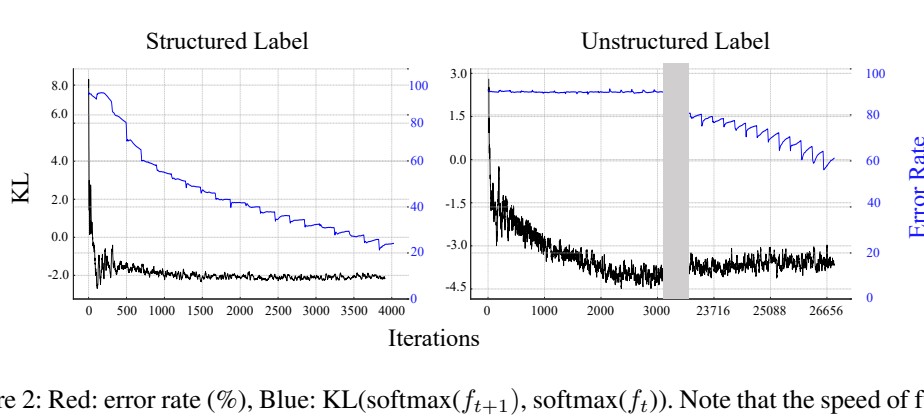

Figure 2: Red: error rate (%), Blue: KL(softmax($f_{t+1}$), softmax($f_t$)). Note that the speed of function $\|\frac{df}{dt}\|$ (KL divergence) decreases much faster than the loss (error), which means the adaptation of NTK to the dataset. For better visualization, we plotted the KL values on a log scale and applied a moving average for both KL and error rate.

feature of unstructured label $\mathcal{Y}_u$ is that it makes the gradient updates extremely unstable during the learning process. This instability is largely due to the randomness of unstructured labels in the mini-batch sampling process of SGD. Specifically, when the model learns, SGD selects a random subset of samples from the dataset for training (minibatch) at each step. However, with unstructured labels, since there is no regularity in these sampled data, the correlation between successive gradients, $\tilde{g}_t$ and $\tilde{g}_{t+1}$, becomes very weak.

In other words, in contrast to structured labels, training with unstructured labels results in each minibatch being completely random in structure, leading to a different label pattern every time. As a result, the likelihood of the gradient being updated in a significantly different direction at each time increases. If at one time the gradient $\tilde{g}_t$ pulled the model in a certain direction, the gradient at the next time $\tilde{g}_{t+1}$ could update in a completely unrelated direction. Through this process, SGD fails to maintain consistency in the direction of the gradient over consecutive time steps, and as training progresses, the gradient updates consistently lose coherence in environments with unstructured labels, leading to a very unstable training process.

### 3.3 SWING PHENOMENON

So, does the training process indeed become 'noisy,' as we suspect? To verify this, we plotted the model's predicted categories for training data by binning them. In Fig. 1, we confirmed that our hypothesis was correct, and observed a phenomenon even more extreme than we had anticipated. At each iteration $t$, the function $f_t$ made completely different predictions on the training dataset, with the bin values consistently concentrated on a specific index different for each $t$. This phenomenon occurred in both models trained with unstructured and structured labels during the very early stages of training, but lasted much longer in the unstructured setting. We call this phenomenon the 'swing phenomenon'.

**Understanding swing phenomenon through NTK**  We can explain this phenomenon through neural tangent kernel (NTK), a functional analysis tool for deep learning.

Consider a loss function, $\mathcal{L}$, made up of $N$ sample-specific losses, $\ell$:

$$\mathcal{L}(\theta) = \frac{1}{N} \sum_{i=1}^{N} \ell(f(x^{(i)}; \theta), y^{(i)}), \tag{5}$$

where $\theta \in \mathbb{R}^P$ is the parameter vector and $f \in \mathbb{R}^C$ outputs the logit vector for a $C$-class classification problem.

Using the gradient flow, $\frac{d\theta}{dt} = -\nabla_\theta \mathcal{L}(\theta) = -\frac{1}{N} \sum_{i=1}^{N} \nabla_\theta f(x^{(i)}; \theta) \nabla_f \ell(f, y^{(i)})$, the temporal change of the function for a specific input, $x$, can be described by the NTK as below:

$$\frac{df(x;\theta)}{dt} = \frac{df(x;\theta)}{d\theta}\frac{d\theta}{dt} = -\frac{1}{N} \sum_{i=1}^{N} \underbrace{\nabla_\theta f(x;\theta)^\top \nabla_\theta f(x^{(i)};\theta)}_{\text{Neural tangent kernel } \Theta} \nabla_f \ell(f, y^{(i)}). \quad (6)$$

The swing phenomenon suggests that the output logit distribution at the initial learning phase is indeed concentrated around a certain label, which indicates that $f(x^{(i)}) \simeq f(x^{(j)})$, $\forall i, j$. Because all the data points lie on the same level surface $L_c = \{x \in \mathcal{X} : f(x;\theta_t) = c\}$ at a time, and suddenly move to another level surface $L'_c = \{x \in \mathcal{X} : f(x;\theta_{t+1}) = c'\}$, we conjecture that the gradient would also be similar for different data points, *i.e.* $\nabla_\theta f(x^{(i)};\theta) \simeq \nabla_\theta f(x^{(j)};\theta) \in \mathbb{R}^{P \times C}$, $\forall i, j$, which suggests the positive definiteness of NTK $\Theta(x, x^{(i)}) \in \mathbb{R}^{C \times C}$:

$$\Theta(x, x^{(i)}) := \nabla_\theta f(x;\theta)^\top \nabla_\theta f(x^{(i)}, \theta) \succeq 0 \quad \forall i. \quad (7)$$

In the early stage, most training samples will likely produce predictions different from the ground truth, resulting in large $\ell$ which would produce large $\nabla_f \ell$ (Sutskever et al., 2013). With a positive-definite NTK, this implies that the function will change very rapidly at the early stage of training.

Fig. 1 confirms our explanation. At each iteration, from a binning perspective, the function moves toward an extreme direction, and since the logits always concentrate in this process, $\|\nabla_f \ell(f, y^{(i)})\|$ takes large values for most samples. As the NTK at this stage is positive definite with large eigenvalues, the function moves fast and cannot converge. However, we know that overparameterized models can memorize all structures, $(\mathcal{X}, \mathcal{Y})$. Convergence means that the sequence becomes increasingly stable as it progresses. In other words, the sequence $f_t := f(x;\theta_t)$ is a Cauchy sequence, which implies that the rate of change $\|\frac{df}{dt}\|$, i.e., the learning speed, must gradually decrease. Thus, in Eq. (6), it can be inferred that the term $\Theta$, related to the learning speed, undergoes a conditioning process during training. As the deep learning model learns, $\Theta$ naturally develops so that it has smaller eigenvalues in magnitude. Based on this analysis, we can derive the following conjecture.

### 3.4 CONJECTURE: PAN FOR GOLD

Based on the observations above, we propose the following hypothesis: the generalization of deep learning arises from the process of learning the function itself. This learning process is highly unstable and causes drastic change of the function. SGD naturally resolves these instabilities and as the learning process stabilizes, what we call 'gold features' naturally emerge. In this context, gold features refer to elements that satisfy Eq. (3). During the learning process, the model must be able to distinguish between gradients and noise, and furthermore, it should not be significantly affected by the curvature of the loss function's surface as it traverses it. By retaining gold features while eliminating the remaining characteristics within the function space, the model achieves generalization.

**Unstructured Labels Leave Out More Gold**   According to our assumption, the generalization of deep learning is ultimately a process of leaving only the 'gold features,' and if we simply perturb the model, those 'gold' will be automatically obtained. The remaining question, then, is how can we leave behind more of this gold? We can find the answer directly in Eq. (3). Generalization is the distinction between the stochastic gradient $\tilde{g}$ and the unintended noise $\varepsilon$ from a functional perspective, and the difficulty of this distinction increases drastically when unstructured labels are used. To distinguish $\tilde{g}$, we must understand it, which implies that $\tilde{g}$ must be consistent over time $t$. In the case of unstructured labels, since there is no structure in the data, this consistency is much lower compared to structured labels. Therefore, to enhance this consistency, as much 'gold feature' as possible must be retained. With this approach, we can simply improve the generalization capability of the existing models, as demonstrated in Sec. 4.

### 3.5 ANALYSIS

To support the above 'pan for gold' hypothesis, we have proposed that the function's speed is very fast at the early stages and that loss does not decrease as fast. We suggested that the reduction in loss occurs as the instability is resolved, leading to a decrease in the function's speed, and that this process

itself produces good features. To verify this conjecture, we present the evidence by answering the following questions: 1. Is our assumption about speed correct? 2. Does a good functional descent actually occur? 3. And finally, do 'good' features really remain as a result?

**Empirical evidence on NTK assumption** To verify this assumption, we conducted the experiment in the following manner. In Eq. (6), we argued that the speed or the spectral norm of NTK decreases over time during the data adaptation process. One of the sufficient conditions for the assumption to hold true is: 1. the speed of the function must decrease, and 2. $\nabla_f \ell(f(x), y)$ should not decrease. To this end, we computed the loss value and the speed of $f_t$ over time by measuring the Kullback–Leibler (KL) Divergence of $f_t$ and $f_{t+1}$ by taking softmax outputs of them:

$$\left\| \frac{df}{dt} \right\| \simeq \|f_{t+1} - f_t\| \sim KL(\text{softmax}(f_{t+1}), \text{softmax}(f_t)). \quad (8)$$

Fig. 2 shows that our assumption is true, as the speed of the function $\frac{df}{dt}$ indeed decreases, while, in contrast, the magnitude of the loss did not decrease. With this and utilizing the fact that $\nabla_f \ell(f(\mathcal{X}, \mathcal{Y})) \propto |\ell|$ (Sutskever et al., 2013), we confirm that our assumption in Eq. (7) is valid.

**The condition is met as learning continues** The next thing we need to look at is the gradient direction itself. As shown in Eq. (3), the direction of the gradient needs to be 'special', i.e. it needs to be robust to noise (assumption 2) and flat along its path (assumption 1). Fig. 3 illustrates this indeed is true. We computed the cosine similarity between the gradient at its current position and that of a nearby position, $\cos(\nabla_\theta \mathcal{L}(\theta_t), \nabla_\theta \mathcal{L}(\theta_t + \delta))$. The blue curve was plotted with $\delta$ being the gradient direction i.e. $\delta \propto \nabla_\theta \mathcal{L}(\theta_t)$, while the black curve uses a random direction as $\delta$. In the random direction case, the loss landscape appears noisy regardless of the iteration, while the higher cosine similarity in the gradient direction case shows that it behaves more like a linear model with a smoother loss surface in the gradient direction.

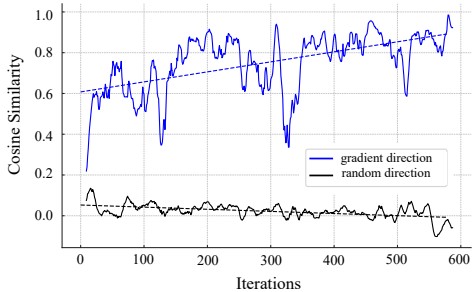

Figure 3: Cosine similarity between the gradients of the current ($\theta_t$) and a nearby point ($\theta_t + \delta$), i.e. $\cos(\nabla_\theta \mathcal{L}(\theta_t), \nabla_\theta \mathcal{L}(\theta_t + \delta))$. Blue: gradient direction ($\delta \propto \nabla_\theta \mathcal{L}(\theta_t)$), Black: random direction ($\delta = $ random). Higher similarity means smaller Hessian in that direction, i.e. smaller $\|\mathcal{H} \cdot \delta\|$.

This shows that even for an unstructured model, Eq. (3) holds, confirming that the model successfully distinguishes between the gradient and noise.

**Learning Actually Occurs** We hypothesize that learning occurs naturally when a 'good' functional gradient descent takes place, distinguishing gradient from noise. This was quantitatively confirmed in Table. 1, while Figure. 4 provides a qualitative validation. We utilized the XAI methodology to see how the model actually identified images, and the results were quite interesting. We wanted to observe how a model trained on unstructured labels would classify images over time by Grad-CAM (Selvaraju et al., 2017). In the early stages when the model just starts to learn, the model tends to focus on random spots for images. However, as learning progresses, it begins focusing on key areas of the object, localizing important features.

## 4 EXPERIMENTS

**Implementation detail** Base on the"pan for gold" hypothesis, we propose "Panning through Unstructured Label (PUL)" to enchance generalization across a wide range of tasks. Our algorithm follows a simple plug-and-play approach, allowing it to be seamlessly applied to any type of dataset. To demonstrate the efficacy of our method, we conducted two types of tasks: 1. unsupervised domain adaptation and 2. object discovery. In both tasks, we assigned unstructured labels to the unlabeled

Train Epoch

Figure 4: Visualization of GradCAM across epochs when training with only unstructured labels. We denote the randomly assigned label as $y_u \in \mathcal{Y}_u$. The first and third rows show GradCAM for the assigned unstructured labels, while second and fourth rows display GradCAM for other labels.

Table 2: SFDA classification accuracy (%) on the Digit datasets. S, M, and U refer to SVHN, MNIST, and USPS, respectively.

|  | $M \to U$ | $S \to M$ | $U \to M$ | Avg |
|---|---|---|---|---|
| Source Only (ERM) | 77.22 | 73.85 | 89.10 | 80.06 |
| Ours | **91.35** | **78.02** | **91.53** | **86.97** |

training data. We added a projector composed of two linear layers after the pre-trained feature extractor and trained the model using cross-entropy loss. To prevent the learned representations from significantly deviating from the source pre-trained representations, we introduced an additional regularization term. Specifically, we applied Kullback-Leibler (KL) divergence between the prediction distributions of the pre-trained model and the model under training. Notably, **we did not use any ground truth labels** from the pre-trained dataset. Instead, we utilized only the pre-trained weights along with the unlabeled training dataset. All experiments were evaluated using the existing codebase (Tim, 2020). In the unsupervised domain adatpation for object detection task, because there was no pre-trained model available for the Cityscapes dataset (Cordts et al., 2016), we utilized both the Cityscapes dataset and the unlabeled Cityscapes-Foggy dataset (Sakaridis et al., 2018) during training to straightforwardly verify the effectiveness of our PUL algorithm. Specifically, we followed the standard object detection training process using the labeled Cityscapes dataset while adopting a method of assigning unstructured labels to the unlabeled data for training.

## 4.1 UNSUPERVISED DOMAIN ADAPTATION

In a narrow sense, generalization refers to the performance gap between a train set and a test set drawn from the same distribution. In this context, the Domain Adaptation task (Ganin et al., 2016; Hoffman et al., 2018; Sun & Saenko, 2016) is an extension of that concept. If a model works well in its target domain despite differences in data distributions across domains, we can say that it has the ability to generalize in a broader sense.

Table 3: SFDA classification accuracy (%) on the Office Home datasets. Ar, Cl, Pr, and Re correspond to the domains defined within the dataset: 'Art', 'Clipart', 'Product', and 'Real-world'.

|  | ar→cl | ar→pr | ar→re | cl→ar | cl→pr | cl→re | pr→ar | pr→cl | pr→re | re→ar | re→cl | re→pr | avg |
|---|---|---|---|---|---|---|---|---|---|---|---|---|---|
| Source Only(ERM) | 44.05 | **65.88** | **73.93** | 52.31 | 61.23 | 64.2 | 51.54 | 39.62 | 72.49 | 64.39 | 45.47 | 77.29 | 59.37 |
| Ours | **45.70** | 65.26 | 72.78 | **55.50** | **63.00** | **65.89** | **53.59** | **42.34** | **72.62** | **65.69** | **48.95** | **77.58** | **60.74** |

In this work, we show that we can effectively tackle one of the challenging tasks in domain adaptation, source-free domain adaptation (SFDA), by utilizing unstructured labels. SFDA (Liang et al., 2020; Ding et al., 2022), a sub-category in unsupervised domain adaptation, aims to improve performance on a target domain given a source pretrained model and an unlabeled target dataset.

Table 2 and 3 compare the performance of our PUL algorithm against the source-only baseline on SFDA tasks for two well-known benchmarks: Digit (Hull, 1994; Lecun et al., 1998; Netzer et al., 2011) and Office-Home (Venkateswara et al., 2017). For a fair comparision, we used LeNet5 (Lecun et al., 1998) for USPS $\leftrightarrow$ MNIST, a modified version of the LeNet5 for SVHN $\rightarrow$ MNIST and ResNet50 (He et al., 2016) for Office-Home, following (Liang et al., 2020). In the experiments, we assigned unstructured labels to three random classes and conducted training for only 2 epochs. When applying KL Divergence, we used a temperature of 4. The result of unsupervised domain adaptation for the object detection task is also provided in Table 4.

As shown in the results, simply leveraging unstructured labels during training effectively addresses the domain gap. We can see that training with unstructured labels naturally eliminates noisy features that do not contribute to reduce the domain gap, retaining the 'gold features' that enhance the generalization performance.

Table 4: The results of unsupervised domain adaptation for object detection on Cityscapes $\rightarrow$ Cityscapes-Foggy.

| Metric | Ours | Base |
|---|---|---|
| MAP 100:95 | **0.207** | 0.194 |
| MAP 50 | **0.331** | 0.324 |

### 4.2 Object Discovery

Object discovery (Siméoni et al., 2023; Shin et al., 2022) refers to the task of automatically identifying objects in images or visual data that are not explicitly labeled. Recently, methods utilizing the attention maps of well-trained deep learning models have become the primary approach for this task (Siméoni et al., 2021). In such cases, the model must be able to accurately detect objects and reflect them into the attention map, even under various environments and conditions. Therefore, the model's generalization ability plays a crucial role in the performance of object discovery.

Table 5 presents the object discovery performance on various trained models on ResNet50, including ImageNet pretrained, DINO (Caron et al., 2021a), and ImageNet pretrained weights further trained using our method. As can be seen from the results, even with just a three epochs of training using unstructured labels, performance can be easily improved.

To visually observe the role of using unstructured labels, we extracted degree maps used for identifying the object, as shown in Fig. 5. As can be seen in the figure, our algorithm removes irrelevant features and helps the model focus more effectively on the object in the image. This suggests that, as discussed in the earlier analysis, our algorithm perturbs the functional space naturally eliminating noisy features while retaining only the gold features that enhance generalization.

Table 5: Accuracies (%) of object discovery using the LOST (Siméoni et al., 2021) with various initial weights. We observed that utilizing unstructured labels during training significantly improves performance in object discovery tasks.

| | VOC07 | VOC12 | COCO20K |
|---|---|---|---|
| Imagenet | 33.83 | 39.06 | 25.50 |
| DINO | 36.84 | 42.67 | 26.47 |
| Ours (PUL) | **37.22** | **43.01** | **27.30** |

### 4.3 Vision Transformers Need Randomness: Unstructured Labels Mitigate the Need For Registers for Quantization

Fig. 5 illustrates another potential of our PUL algorithm in improving quantization. As mentioned in Sec. 3.4, PUL induces the model to favor a specific function space by artificially 'swinging' within function space and relaxing its tension. As a result, we observed that artifacts, such as outliers in the attention map, were reduced. This aligns with physical intuition, where relaxation of tension leads to the stabilization of the space, and high-energy regions, like artifacts, are naturally eliminated.

This characteristic not only enhances object discovery by improving attention maps but also helps address quantization issues caused by massive activation in large models. Massive activation, which occurs when certain neurons become excessively large, disrupts the quantization process by reducing precision, thereby degrading model performance. This issue is particularly problematic because it significantly affects the quantization performance of large models. To address this, large lan-

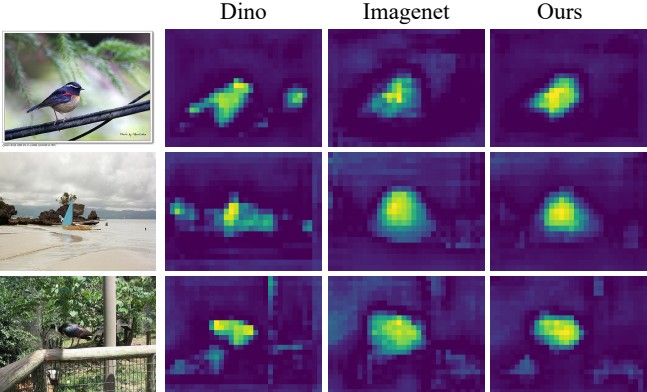

Figure 5: Visualization of degree maps (Siméoni et al., 2021) for various initialization weights. Notably, our algorithm guides the model to focus more on the objects within the image.

guage models (LLMs) adopt massive-activation-aware quantization algorithms (Xiao et al., 2024; Yao et al., 2022a;b). In vision models, the problem is mitigated by adding a "register" token, which increases sequence length and helps reduce these artifacts (Darcet et al., 2023). Fig. 6 demonstrates that the PUL algorithm offers a simpler solution. By applying the PUL algorithm, the instability caused by massive activation is alleviated, resulting in a more balanced distribution of activations.

### 4.4 EDGE-LIKE BEHAVIOR OF SALIENCY MAPS ACTUALLY MEANS GOOD

One of the main criticisms of eXplainable Artificial Intelligence (XAI) methodologies is that saliency maps of existing methods like GradCAM (Selvaraju et al., 2017) and GradCAM++ (Chattopadhay et al., 2018) are fundamentally edge-concentrating algorithm (Adebayo et al., 2018). In other words, these saliency methods often end up highlighting areas with concentrated edges, such as image contours, which doesn't differ significantly from simply focusing on edges, rather than providing a deeper understanding of the AI model's decision-making process. This raises questions about the reliability of XAI methodologies. However, our research shows that these edge concentrated explanations actually reflect the model's learning process and judgment capabilities.

As shown in Fig. 4, this tendency actually reflects the model's learning process and its judgment capabilities. Fig. 4 shows that there exists no edge concentration bias at the start. However, the bias gradually emerges over time as learning progresses. Furthermore, this tendency appears **only for the target label** and not for other labels. Based on this, we can conclude that the edge-bias phenomenon is not a weakness, but instead a natural generalization process that deep learning models acquire as they continue to learn.

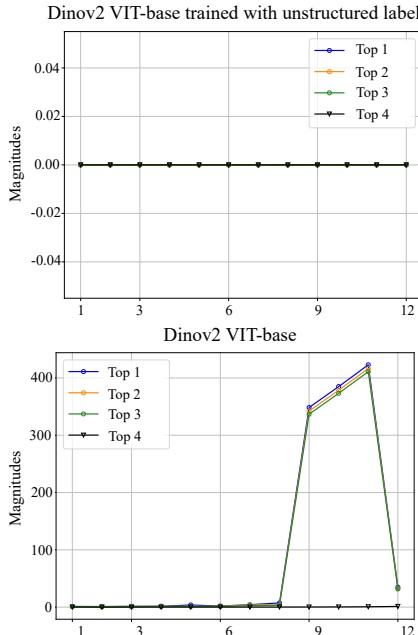

Figure 6: Layer-wise magnitude of two different models. We plotted top 4 activation outliers per layer (x-axis). With PUL, the massive activation diminishes.

## 5 CONCLUSION

In this paper, we present a provocative claim that the process of "learning from data" occurs independently of human-imposed structures. To support this, we introduce the bold alternative hypothesis called the "Pan for Gold". Through extensive experiments, we quantitatively and qualitatively demonstrate the validity of our hypothesis. Based on these findings, we introduce our algorithm, Panning through Unstructured Label (PUL), showing that it can improve performance across various fields with a simple approach, while also providing a fresh reinterpretation of existing beliefs.

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
