# OpenReview forum: "Pan for gold"
_ICLR.cc/2025/Conference — ICLR 2025 Conference Withdrawn Submission_

### Official Review · Reviewer_XxDH · 2024-10-24

**Soundness:** 1
**Presentation:** 1
**Contribution:** 1
**Rating:** 1
**Confidence:** 4

**Summary:**

The paper presents the observation that training on data with random labels can learn useful “features”, and the author suggest doing this as an unsupervised per-training scheme, though the details of how are left very vague. Unfortunately the written quality of the paper is very low, it is difficult to determine much more than this.

**Strengths:**

The paper presents the observation that training on data with random labels as an unsupervised per-training scheme can learn useful “features”, though the details of how are left very vague.
I am not familiar enough with this area of literature to know how novel this claim is. To me it does not seem surprising that training on random labels leads to a better model initialisation than the no pertaining.
Unfortunately the written quality of the paper is very low and thus it is difficult to determine if the paper really offers any strengths.

**Weaknesses:**

I apologise if English is not the first language of the authors but the written quality of the paper is sadly unacceptable for ICLR. At the moment the paper is almost impossible to follow. While there are a lot of machine learning terms mention the sentences are so unspecific and vague, lacking in concrete definitions and jumping from place to place it makes assessing the ideas of the authors impossible for me. I do not know if LLM’s have been used in the writing of the paper, or in translating from another language but the final quality is just not good enough at the moment.

The issues with the write are as follows:

1. Use of vague terms, which are not defined. For example “structure”, “function speed”, “Noisy features”, “Gold features”

2. Vague handy wavey claims not back up with references: “This aligns with physical intuition, where relaxation of tension leads to the stabilization of the space, and high-energy regions, like artifacts, are naturally eliminated.”

3. Reference to a “Panning through Unstructured Label (PUL) algorithm” which is never really defined, you have to try and glean what the algorithm is from passing comments.

4. The details of the experiment are extremely vague. For example. The full text exampling an experiment is: “Table 5 presents the object discovery performance on various trained models on ResNet50, including ImageNet pretrained, DINO (Caron et al., 2021a), and ImageNet pretrained weights further trained using our method. As can be seen from the results, even with just a three epochs of training using unstructured labels, performance can be easily improved. ”

5. Baselines for the experiments are not explained, the experiments jump from place to place with very little intuition and justification of why important choices were made We would like to recommend the paper is rewritten with a greater leave of specificity, intuition and detail before being resubmitted.


Other.

The experiments are extremely limited, typically only considering a single model, data set with no mention of repeats or the training procedure
Baselines for the experiments are extremely limited a single baseline is used, with little detail of what this baseline was and why it should lead to a fair comparison
In the conclusion the authors claim they introduce a “bold alternative hypothesis called the “Pan for Gold”,” but after reading the paper I’m am left wondering what this current hypothesis that “Pan for Gold” is mean to be an alternative too? This is never explained in any level of detail.

**Questions:**

see above

**Details Of Ethics Concerns:**

The paper is horribly written and throws at the face of the reader, all throughout, ill-defined terms high level philosophical terms that I do not know what to make of.

---

### Official Review · Reviewer_N14h · 2024-11-03

**Soundness:** 2
**Presentation:** 2
**Contribution:** 2
**Rating:** 3
**Confidence:** 3

**Summary:**

This paper investigates the generalization of deep learning methods when faced with random labels. It argues that generalization is not about learning the structure of data (X, Y), but rather follows a stochastic process that initially fluctuates before converging to a stable function space, akin to "panning for gold." Experiments on unsupervised domain adaptation show that using random labels with KL regularization outperforms the source-only baseline, which does not apply any adaptation. Exploratory analysis also shows that the proposed random labels reduce outliers in the attention map, resulting in a more balanced distribution that is more suitable for quantization.

**Strengths:**

The paper is well-written and presents an interesting analysis of generalization through stochasticity. I like the idea of decoupling the impact of human-imposed labels in generalization.

The paper carries out principled analysis on neural tangent kernel that demonstrates the swing phonemenon in the learning process. The visualizations of learning process through gradCAM and analysis on saliency maps are interesting.

**Weaknesses:**

Although I liked the exploration of stochasticity and its role in learning, I think the paper places too much emphasis on supervised data and stochasticity while neglecting other important factors in unsupervised and semi-supervised learning. The experiments are relatively weak because they rely on a supervised model trained with ground-truth data and KL regularization, which does not fully demonstrate the impact of stochasticity.

**1. Definition of structure, the role of supervised labels, and considerations for unsupervised/semi-supervised learning**

The paper frequently mentions "structure" but does not provide a clear definition. It appears to me that "structure" refers to supervised class labels assigned by humans, and the paper argues that the "structure" itself is not the essence of the 'gold' result and that learning also happens with random labels.

I disagree with the notion that "the goal of deep learning is to learn from data according to structures defined by humans." I think the supervised signal is only one source of information that models use to learn. Other sources of information include data itself as used in unsupervised or self-supervised learning, information from a model decision boundary manifested through unlabeled data as in transductive and semi-supervised learning setups, and assumptions about the world such as convolution for image processing. The paper solely focuses on the structure from supervisory signals and ignores other sources of information, which also plays a crucial role in learning that could be attributed to stochasticity in this paper.

**2. Experiments**

The experiments focus on domain adaptation where a supervised model has been trained on ground-truth labels in the source domain, and the goal is to adapt the model to a closely related target domain. This setup weakens the empirical results because the model is initialized with a learned representation from ground-truth labels and does not fully demonstrate the impact of stochasticity in a learning-from-scratch setup. Moreover, the model only performs small adjustments due to the constraint of KL regularization that penalizes the model for deviating from the source model. It is well-known that, in semi-supervised learning, a model could outperform the source-only baseline without any target labels. It is unclear if the improves is from stochasticity or semi-supervised learning (clustering assumption, KL, transduction through batchnorm).

**Questions:**

1. Could you provide the definition of structure in this paper?
2. How do you distinguish the impact of stochasticity with the impact of unsupervised/semi-supervised representation learning/model-inductive-bias/compression?
3. In Table one, "we applied transfer learning to the frozen encoder." Could you elaborate how is transfer learning carried out?

Minor:
In Fig 2, "red" should be "black"

---

### Official Review · Reviewer_uKV2 · 2024-11-03

**Soundness:** 2
**Presentation:** 3
**Contribution:** 2
**Rating:** 3
**Confidence:** 4

**Summary:**

This paper presents a new hypothesis about generalization in deep learning, suggesting it's not about learning structured patterns in data but rather a "Pan for Gold" process where SGD naturally filters useful features, and proposes the PUL algorithm utilizing random labels, demonstrating performance improvements in domain adaptation and object discovery tasks.

**Strengths:**

1. The paper presents a novel and interesting perspective on deep learning generalization by proposing a new hypothesis about the role of stochasticity in learning meaningful features
2. The proposed methodology is remarkably simple yet demonstrates effectiveness, requiring only random labels and a few additional training steps
3. The theoretical analysis through Neural Tangent Kernel provides mathematical insights into the learning dynamics and supports the main hypothesis
4. The experimental results show significant performance improvements across various applications including domain adaptation and object discovery tasks, demonstrating the practical utility of the proposed method

**Weaknesses:**

1. The paper critically lacks essential experimental details needed for reproduction, including the specific method of generating unstructured labels, exact model architecture, hyperparameter settings, and detailed training procedures, making it difficult to validate the claims independently.

2. The paper lacks clear explanation about whether unstructured labels remain fixed during training. Based on the paper's content, it appears that labels are fixed, in which case neural networks would inevitably learn visual features in the process of memorizing image-label pairs, as they need to recognize some visual patterns to distinguish between images even with random labels. This suggests that learning meaningful features might be a natural consequence of the memorization process rather than the proposed "Pan for Gold" hypothesis.

3. The theoretical analysis is insufficient as the paper lacks in-depth discussion on why the "Pan for Gold" process leads to good generalization, focusing merely on describing phenomena without explaining the underlying mechanisms

4. The experimental validation is limited, lacking analysis of performance with longer training epochs in Sections 4.1 and 4.2, missing ablation studies on the number of unstructured labels, and failing to provide sensitivity analysis for various hyperparameters.

**Questions:**

1. Are the unstructured labels fixed during training or regenerated each epoch? Have you also experimented with changing labels at each epoch? This would be an important experiment as it could lead to completely different learning dynamics since the network cannot memorize stable image-label pairs.

2. In Sections 4.1 and 4.2, how does the performance change with longer training periods? The paper only shows results with 2-3 epochs, but longer training analysis is necessary to understand the stability and effectiveness of the method.

3. How was the optimal number of random classes determined in the PUL algorithm? Did you perform any experiments with different numbers of classes?

---

### Official Review · Reviewer_wyxn · 2024-11-03

**Soundness:** 1
**Presentation:** 2
**Contribution:** 2
**Rating:** 3
**Confidence:** 4

**Summary:**

This paper introduces the "Pan for Gold" hypothesis, which challenges the traditional view that structured labels and well-defined loss functions are essential for deep learning models to learn meaningful representations and generalize well. The authors propose that generalization emerges naturally through the stochasticity inherent in SGD when training overparameterized models, even with unstructured (randomized) labels. They suggest that SGD acts like panning for gold, where valuable features are naturally sifted out from noise without relying on human-imposed structures.
To support this hypothesis, the authors conduct experiments where models are trained on datasets with completely randomized labels. Surprisingly, these models still learn meaningful features, as evidenced by improved performance over random initialization. They analyze this phenomenon using the NTK framework and observe a "swing phenomenon," where model outputs fluctuate significantly during early training stages.
Based on these observations, they introduce the PUL algorithm. They demonstrate PUL's effectiveness in tasks like unsupervised domain adaptation and object discovery. Additionally, they suggest that PUL mitigates issues like massive activation in vision transformers, aiding in model quantization.

**Strengths:**

* **Challenging Conventional Wisdom:** The paper attempts to question the traditional beliefs about the necessity of structured labels and loss functions in deep learning, which is an interesting and bold endeavor.

* **Novel Hypothesis Introduction:** The "Pan for Gold" hypothesis is a creative metaphor that could inspire new ways of thinking about generalization in deep learning.

* **Exploration of Unstructured Labels:** Investigating the effects of training with unstructured labels is an intresting approach that could uncover overlooked aspects of model training dynamics.

**Weaknesses:**

* **Lack of Theoretical Rigor:** The paper makes strong claims without providing a solid theoretical foundation. The mathematical analysis is superficial and does not rigorously justify the "Pan for Gold" hypothesis or explain why unstructured labels should lead to better generalization.
* **Insufficient Empirical Evidence:** The experimental evaluation is limited and inadequate to support the bold claims made. Experiments are conducted on small datasets like MNIST, CIFAR-10 and SVHN, which are not representative of modern large-scale tasks. The performance improvements reported are marginal and could be due to experimental noise.
* **No Comparison with Baselines:** The paper fails to compare the proposed PUL algorithm with established baselines or state-of-the-art methods in the respective tasks. Without such comparisons, it's impossible to assess the significance of the results or attribute improvements to the proposed method.
* **Overgeneralization of Findings:** The authors make sweeping generalizations about deep learning based on limited and specific experiments. The claim that generalization emerges naturally through SGD in overparameterized models trained with unstructured labels is not convincingly demonstrated.
Methodological Issues: Key details about the experimental setup are missing or unclear, hindering reproducibility. For example, the process of assigning unstructured labels, hyperparameter settings, and specifics of the PUL algorithm are not adequately described.

Weak Analysis of Results: The paper lacks a thorough analysis of the results. It does not explore alternative explanations for the observed phenomena or consider confounding factors. The interpretations often rely on anecdotal observations rather than rigorous investigation.
Ambiguous Writing and Clarity Issues: The paper is difficult to follow in several sections due to ambiguous explanations and poor organization. Key concepts are not clearly defined, and the narrative lacks coherence, making it challenging to understand the proposed ideas fully.

**Questions:**

1. **Theoretical Justification:** Can the authors provide a rigorous theoretical framework to support the "Pan for Gold" hypothesis? Specifically, how does SGD with unstructured labels in overparameterized models lead to meaningful generalization, and what are the underlying mechanisms?
2. **Experimental Validation on Larger Datasets:** Have the authors considered testing the PUL algorithm on larger and more diverse datasets to validate the generality of their claims? Small-scale datasets may not capture the complexities of modern deep-learning tasks.
3. **Comparative Analysis with Baselines:** How does the PUL algorithm perform compared to existing state-of-the-art methods in unsupervised domain adaptation and object discovery?
4. **Clarity on Pan for Gold Hypothesis:** The hypothesis seems to conflate the effects of noise and regularization in SGD with meaningful learning from unstructured data. Can the authors clarify how their hypothesis differs from existing theories on overparameterization and implicit regularization?

---

### Official Review · Reviewer_qcaF · 2024-11-04

**Soundness:** 1
**Presentation:** 2
**Contribution:** 1
**Rating:** 1
**Confidence:** 4

**Summary:**

This paper explores the finding that training neural networks with random labels leads to substantial performance improvements in comparison to randomly initialised neural networks. Authors claim that these experimental findings (supported by empirical evidence & overlapping with prior work -- see below) highlight that the process of learning from data occurs independently of human-imposed structure and inform a novel perspective on the way neural networks work and do not discuss the work's limitations. Authors go on to proposing the use of random labels to fine-tuned pre-trained backbones to improve downstream generalisation.

**Strengths:**

- clear and well-written: the manuscript is well-written and easy to follow
- relevant topic: the authors tackle an interesting finding (i.e., training with random labels leads to substantial performance increase) that is relevant to the community and connected to multiple popular topics like self-supervised representation learning as well as the emerging idea of a universal representation

[1] Bojanowski, Piotr, and Armand Joulin. "Unsupervised learning by predicting noise." International Conference on Machine Learning. PMLR, 2017.
[2] Reizinger, Patrik, et al. "Cross-Entropy Is All You Need To Invert the Data Generating Process." arXiv preprint arXiv:2410.21869 (2024).
[3] Huh, Minyoung, et al. "The platonic representation hypothesis." arXiv preprint arXiv:2405.07987 (2024).

**Weaknesses:**

- strong overlap with non-cited work/lack of novelty: the authors centered their work around the observation that random labels offers substantial performance improvement which they claim is a novel finding ("We completely removed the structure from the learning process by randomizing the class labels, and found that the model actually was able to learn from data despite the complete randomization and even performed better from a generalization perspective."). In fact, this observation has been presented and discussed in several works in the past, including [1], which is not cited by the authors.
- soundness of claims: the paper makes bold claims about "how neural networks learn" and what drives this process ("we present as provocative claim that the process of learning from data happens independently of human-imposed structures. To support this, we introduce the bold alternative hypothesis called the “Pan for Gold”. ").  These claims remain conjectures and hypothesis which are only supported by empirical evidence that the network learns from random labels which does not prove the author's "pan for gold" hypothesis. Additionally, authors further justify the relevance of their work by relying on GradCam visualisation, a method proven to be unreliable -- as also mentioned by authors.
- confidence in empirical findings: while the paper is well-written and clear, there is a lack of polishing of figures and of empirical results which impedes clarity and well as confidence in empirical results (e.g., missing axis labels, randomly masked out portions of curves, single seed experiments, core findings in section one are conducted on two small scale datasets and a single architecture type).
- missing sections: the authors omit important sections to their work including a related work section and a discussion of the paper's limitations.


[1] Bojanowski, Piotr, and Armand Joulin. "Unsupervised learning by predicting noise." International Conference on Machine Learning. PMLR, 2017.

**Questions:**

- can authors discuss the work's limitations and potential impact on future work?
- can authors discuss potential overlap with prior work notably [1] (see above)?
- can authors adjust figure 1 with axis labels and explain why the number of samples (if I understand correctly) varies between epoch 1 and 5?

---

### Note · Authors · 2024-11-12

I have read and agree with the venue's withdrawal policy on behalf of myself and my co-authors.